

**Large-scale predictions of saltmarsh carbon stock based on simple observations of plant**
**community and soil type.**
Hilary Ford[1,2,], Angus Garbutt[3], Mollie Duggan-Edwards[1], Jordi F. Pagès[1], Rachel Harvey[3], Cai
Ladd[1,4], Martin W. Skov[1]
[1] *School of Ocean Sciences, Bangor University, Anglesey, LL59 5AB, United Kingdom*
[2] *School of Environment, Natural Resources and Geography, Bangor University, Bangor, LL57*
*2DG, United Kingdom*
[3] *Centre for Ecology and Hydrology, Environment Centre Wales, Bangor, LL57 2UW, United*
*Kingdom*
[4] *Department of Geography, Swansea University, Wallace Building, Singleton Park, Swansea*
*SA2 8PP, United Kingdom*
*Correspondence to*: Hilary Ford (hilary.ford@bangor.ac.uk)



**Abstract.**
Carbon stored in coastal wetland ecosystems is of global relevance to climate regulation.
Broad-scale inventories of this 'blue' carbon store are currently lacking and labour intensive.
Sampling 23 salt marshes in the United Kingdom, we developed a Saltmarsh Carbon Stock
Predictor (SCSP) with the capacity to predict up to 44% of spatial variation in soil organic
carbon (SOC) from simple observations of plant community and soil type. Classification of
soils into two types (sandy or not-sandy) explained 32% of variation in SOC. Plant community
type (5 vegetation classes) explained 37% of variation. Combined information on soil and
plant community types explained 44% of variation in SOC. GIS maps of SOC were produced
for all salt marshes in Wales (~4000 hectares), using existing soil maps and governmental
vegetation data, demonstrating the application of the SCSP for large-scale predictions of blue
carbon stores and the use of plant community traits for predicting ecosystem services.



## 1 Introduction

Implementation of environmental policy and management via 'the ecosystem approach'

requires a broad-scale knowledge of the distribution of natural stocks and ecosystem services

(McKenzie et al., 2014; Meiner et al., 2013; TEEB, 2010; UK National Ecosystem Assessment,

2014). Spatial information is often patchy and for some ecosystem stocks and services it is

almost entirely lacking. The 'predictive tool' approach, based on mathematical modelling, was

traditionally used in population and resource distributional mapping (Cuddington et al.,

2013), and has recently been applied to the predictive mapping of ecosystem services

(McHenry et al., 2017). Significant advances have been made in predicting ecosystem service

provision in terrestrial systems, such as agricultural landscapes, freshwater habitats and

forests (Ding and Nunes, 2014; Emmett et al., 2016; Vigerstol and Aukema, 2011). In contrast,

there are few predictive tools for coastal systems which, combined with a shortage of baseline

data for many environmental variables (Robins et al., 2016), means that distributional maps

of ecosystem services and stocks are lacking for global coastlines (Meiner et al., 2013).

Coastal wetlands (mangroves, tidal marshes and seagrasses) sequester significant amounts of

'blue carbon', which they retain in long-lived, primarily below-ground, soil organic carbon

(SOC) stores (Howard et al., 2017; Luisetti et al., 2013). Global strategies for integrating blue

carbon storage into greenhouse-gas accounting have been proposed (IPCC, 2014). However,

a global inventory of blue carbon remains a challenge, as empirical observations of SOC stocks

in coastal wetlands are expensive, scarce and unevenly distributed, with few records even for

relatively well-studied areas such as Europe (Beaumont et al., 2014). Ecosystem service maps

for the UK National Ecosystem Assessment (NEA) for Wales, the focal region of the present

study, characterised salt marshes as the lowest category of carbon storage relative to all other



terrestrial habitats (Scholefield, 2013). SOC stocks in Welsh salt marshes may be under-
estimated due to incomplete habitat mapping of inter-tidal areas. Rolling out empirical
observations of below-ground SOC stock across large scales of blue carbon systems is not a
practicable and affordable short-term solution to the lag between management ambition and
carbon inventorying. Predictive mapping of carbon stocks holds great promise; it has been
extensively trialled for terrestrial systems (Emmett et al., 2016; Gray et al., 2013; Rossel et al.,
2014), but rarely applied to blue carbon ecosystems (Gress et al., 2017; Meiner et al., 2013).
Predictive models of ecosystem services typically use a combination of predictor variables
(Posner et al., 2016). For carbon storage, predictors such as climate, soil type, sedimentary
classification and habitat or land management type are commonly used (Chaplin-Kramer et
al., 2015; Jardine and Siikamäki, 2014; Kelleway et al., 2016). Many ecosystem service models
that include carbon storage predictions are computationally sophisticated, operationally time
consuming and require specialists for their operation and interpretation (Posner et al., 2016),
all of which reduces the scope for their use by landscape managers. Simple predictive tools
that incorporate readily available spatial information with ground-truthed field
measurements might be a more attractive option for use in the field. For example, a recent
study by Emmett et al. (2016) proposed soil pH as a potential metric for ecosystem service
provision, at catchment scale, accounting for 45% of variation in ecosystem service supply.
Recent work has explicitly linked SOC stock to both soil properties and plant community
parameters for terrestrial and coastal grasslands (Bai et al., 2016; Manning et al., 2015). In
addition, these SOC stores are further mediated by climatic factors (e.g. precipitation), and
land-use management (e.g. livestock grazing intensity) (Ford et al., 2012; Tanentzap and
Coomes, 2012; Yang et al., 2010). Classification of soils by texture can be useful for quantifying



soil organic matter (SOM) content and therefore indicating SOC stock (O'Brien et al., 2015).
In particular, a strong positive correlation between clay content and SOC stock is apparent
due to the adsorption of organics to clay particles (Arrouays et al., 2006; Hassink, 1997; Oades,
1988). The composition of the plant community, presence of dominant species and plant
diversity largely determine root properties (e.g. biomass, turnover and exudates), which
further influence SOM content and SOC stock (De Deyn et al., 2008; Ford et al., 2016). Species-
rich plant communities are also often functionally diverse, with differing root strategies
leading to enhanced root biomass (Loreau et al., 2001) and consequent impacts on SOC stock
(Jones and Donnelly, 2004). Moreover, particular life history strategies or plant traits can also
be associated with enhanced carbon capture and storage, for example fast growth rates or
the production of recalcitrant litter that is slow to break down (Yapp et al., 2010).
The ability to easily and quickly predict saltmarsh SOC stock from plant community
assemblages and / or soil type would provide the potential to update the current inventory
of blue carbon on a regional, biogeographical or national scale. This would be of interest to a
wide group of stake-holders including academics, the IPCC, the Blue Carbon Initiative
(http://thebluecarboninitiative.org/) and governmental / non-governmental land managers.
Here we present a range of predictive models for SOC stock based on plant (vegetation type,
class, species richness, root biomass) and soil (simplified type or texture category) parameters
measured across 23 salt marshes in Wales, UK. In addition, we used a subset of these models
to create a novel tool for practitioners – the Saltmarsh Carbon Stock Predictor (SCSP) - for
predicting    and    mapping    the    SOC    stock    of    Welsh    salt    marshes
(https://www.saltmarshapp.com/saltmarsh-tool/); alongside a simplified version designed
for use by the general public - the Saltmarsh App (https://www.saltmarshapp.com/).



## 2 Materials and methods

2.1. Site selection

Twenty-three saltmarsh sites were sampled for vegetation and soil properties in July 2015: 10 in north or mid Wales and 13 in south Wales, UK (Fig.1) representing a range of marsh typologies. The Severn estuary in the south-east was excluded due to nesting bird restrictions. We used the British National Vegetation Classification (NVC) scheme to characterise vegetation communities (Rodwell, 2000). Enabling us to make our 'quadrat-scale' results comparable to existing national NVC maps, thereby allowing estimates of SOC stocks to be up-scaled across all Welsh marshes (see section *2.5.*). Unpublished work also indicated a link between NVC and SOC in saltmarsh habitats (Kingham, 2013). Four of the most common vegetation types (= 5 NVC classes) were assessed in this study (Table 1); they were chosen as they are widespread and common the UK, (Table 1) and present at all study sites according to governmental (Natural Resources Wales, NRW) NVC maps (e.g. Fig. S1, Supplement). At each study site, four 1 x 1 m quadrats were sampled per vegetation type (each quadrat ca. 10 metres apart along a transect line). In some specific locations, where extent was limited, only two quadrats per vegetation type were assessed. Note that the 4 vegetation types equate to 5 NVC classes as the *Juncus maritimus* community is divided into two distinct classes (Table 1).

2.2. Plant community and root biomass

Above-ground vegetation characteristics were measured within each $1 \times 1$ m quadrat. Percentage cover of each plant species was estimated by eye. Plant species richness was recorded as the number of species present per quadrat. Shannon-Weiner index [S-W index



(H')] was calculated as a measure of plant diversity based on species cover. NVC classes
associated with each vegetation type (Table 1) were verified for each quadrat using the
Tablefit v1.1 software (Hill, 2011). Root dry biomass was determined for 0 - 10 cm depth using
a 2.6 cm diameter corer: roots were removed from sediment, washed and then dried at 60°C
for 72 hours. All plant nomenclature followed Stace (2010).
2.3. Soil characteristics, SOC stock and field texture test
Soil characteristics were measured from within each 1 x 1 m quadrat. Soil samples, of ~ 10 g
(fresh mass) from the top 10 cm, were taken from within each quadrat, diluted to a ratio of
1:2.5 by volume with deionised water and measured for electrical conductivity (EC) and pH
(*Jenway* 4320 conductivity meter, *Hanna* pH209 pH meter). EC was used as a proxy for salinity.
Soil bulk density samples were taken using a stainless-steel ring (3.1 cm height, 7.5 cm
diameter) inserted vertically into the soil (from a depth of 2 cm to 9.5 cm deep) to quantify
the top 10 cm of soil. Samples were dried at 105 °C for 72 hours to assess soil moisture content
and soil bulk density. The dried samples were ground and sub-sampled for loss-on-ignition
analysis (375 °C, 16 h) to estimate SOM content (Ball, 1964). SOC stock was calculated from
bulk density and SOM with SOC content estimated as 55 % of SOM (Emmett et al., 2010).
Root-free soil samples (1 per quadrat at 5 cm depth) were classified into 12 soil texture
categories using the British Columbia protocol for estimating soil texture in the field
(https://www.for.gov.bc.ca/isb/forms/lib/fs238.pdf) based on graininess, moistness,
stickiness and ability to hold a form without breaking apart when rolled. Soil was also assigned
a simplified soil type of 'Sandy' or 'Non-sandy' (Table 2). These approaches were chosen over
conventional soil grain-size assessment as they facilitate inexpensive, broad-scale
observations where soils can be classified by non-experts in a few minutes in the field.





**Table 1**. Saltmarsh vegetation types, associated National Vegetation Classification (NVC) class
and marsh intertidal position (zone).

| Vegetation type | NVC class | Marsh zone |
|---|---|---|
| *Puccinellia maritima* community | SM13 | Low / mid |
| *Atriplex portulacoides* community | SM14 | Mid / high |
| *Juncus gerardii* community | SM16 | Mid / high |
| *Juncus maritimus* community | SM15 | Mid / high |
| "        "        " | SM18 | Mid / high |

*NB J. maritimus* community is divided into two NVC classes
**Table 2.** Soil texture categories [British Columbia protocol for estimating soil texture in the
field (https://www.for.gov.bc.ca/isb/forms/lib/fs238.pdf)] and simplified soil type.

| Soil texture category | | Soil category description | Simplified soil type |
|---|---|---|---|
| S | Sand | 85 - 100 % sand | Sandy |
| SL | Sandy loam | 45 - 80 % sand | Sandy |
| FSL | Fine sandy loam | 46 – 80 % fine sandy | Sandy |
| SC | Sandy clay | 45 - 65 % clay | Sandy |
| Si | Silt | 0 - 20 % sand | Non-sandy |
| SiL | Silt loam | 0 - 50 % sand | Non-sandy |
| L | Loam | 20 - 50 % sand | Non-sandy |
| CL | Clay loam | 20 - 45 % sand | Non-sandy |
| SiCL | Silty clay loam | 0 - 20 % sand | Non-sandy |
| SiC | Silty clay | 0 - 20 % sand | Non-sandy |
| C | Clay | > 40 % clay (0 - 45 % sand) | Non-sandy |
| O | Organic | > 30 % OM | Non-sandy |







Figure 1. The 23 Welsh salt marshes included in the study.




2.4. Analysis: Explanatory variables and prediction of SOC stock
The relationship between the response variable 'SOC stock' and the explanatory variables was
determined using uni- or bi-variate linear mixed effects models. This was done in order to
keep the models as simple as possible, to be able to scale the results up to the landscape-
scale using available GIS layers (see subsection *2.6*) and with the final aim of being of direct
use for practitioners. The explanatory variables we entered in the models were the fixed
categorical variables 'vegetation type' (4 levels: *P. maritima* community, *A. portulacoides*
community, *J. gerardii* community, *J. maritimus* community), 'NVC class' (5 levels: SM13,
SM14, SM16, SM15, SM18), 'simplified soil type' (2 levels : sandy, non-sandy), 'soil texture'
(12 levels: sand, sandy loam, fine sandy loam, sandy clay, silt, silt loam, loam, clay loam, silty
clay loam, silty clay, clay, organic) and the continuous variables 'root biomass' and 'plant
species richness'. The categorical variable 'vegetation type' was nested within the random
effects 'saltmarsh site' (23 levels: e.g. Morfa Harlech) and 'location' (2 levels: north or south
Wales) (e.g. Carbon_stock ~ Soil_type + NVC, random = ~1|Location/Site/Veg_type).
Inspection of residuals and Bartlett's test detected a clear violation of the assumption of
homoscedasticity. We addressed this issue by adding a constant variance function (varIdent)
as weights into the linear mixed effects models, to take into account the differences in
variance across groups (e.g. vegetation type, NVC class, simplified soil type).  Final models
were selected on the basis of the lowest Akaike's Information Criteria (AIC) (Zuur et al., 2009).
Likelihood-ratio based pseudo R-squared were calculated for final models (Grömping, 2006).
The final uni- and bi-variate models we tested were the following: i) NVC_model ('NVC class'
only); ii) Soil_model ('simplified soil type' only); iii) Veg_soil_model ('vegetation type' and
'simplified soil type' combined); iv) NVC_soil_model ('NVC class' and 'simplified soil type'



combined). SOC stock predictions were calculated from the coefficients of the final linear
mixed effects models. For example, the NVC_soil_model values for each explanatory variable
for coefficient 1 (i.e. simplified soil type: sandy, non-sandy) and coefficient 2 (i.e. NVC class:
SM13, SM14, SM15, SM16, SM18) were summed and added to the model intercept giving a
model prediction of SOC stock for each model in tonnes of carbon per hectare (t C ha$^{-1}$). All
analysis was carried out in R (R Core Team, 2016).
2.5. Model selection justification for the SCSP tool and the Saltmarsh App
The SCSP tool (Skov et al., 2016; https://www.saltmarshapp.com/saltmarsh-tool/) was
designed to be used primarily by expert practitioners whereas the Saltmarsh App
(https://www.saltmarshapp.com/) was aimed at the general public. Therefore the models
they utilise to predict saltmarsh SOC stock differ based on access to data sources. The SCSP
tool offers two types of information: i) a look up table for predicted SOC stock (t C ha$^{-1}$)
provided either NVC class (NVC_model), simplified soil type (Soil_model) or both
(NVC_soil_model) are known; and ii) a GIS map layer and series of maps (see subsection *2.6*).
The NVC_soil_model was used for The SCSP tool as existing governmental maps are already
categorised by NVC class. The carbon calculator component of the Saltmarsh App was based
on the Veg_soil_model. This model was selected as vegetation type was assessed as easier to
determine than NVC class by non-experts (e.g. citizen-scientists) in the field. For both the SCSP
tool and the Saltmarsh app 'simplified soil type' was used instead of 'soil texture category' as
simplified soil type was both easier to assess in the field by non-experts and more
straightforward to map using existing soil maps.





2.6. Scaling-up: SOC Stock mapping
As part of the SCSP tool, a GIS shapefile (referred to as the SCSP shapefile) was developed to
illustrate how information on NVC class and simplified soil types (sandy v non-sandy) can be
integrated into broad-scale mapping of SOC stocks in saltmarshes across Wales, UK. The SCSP
shapefile illustrated SOC stocks for marshes across Wales utilising the predictive power of the
linear mixed effects models obtained in the statistical analyses (section 2.4) for: A) 'NVC class'
only (NVC_model); B) 'Simplified soil type' only (Soil_model); C) 'NVC and simplified soil type'
combined, (NVC_soil_model); D) 'NVC and simplified soil type' combined (NVC_soil_model)
plus predictions based on 'simplified soil type' (Soil_model) where SOC predictions for NVC
pioneer communities were not known. Estimates of the total amount of carbon (t C) for all
marshes visible, for the 'Area' of the saltmarsh (%) for which we had the necessary
information to make predictions were calculated for each map. For example, Laugharne
marsh (Fig. 2) included NVC classes for which the study did not have predictive SOC to NVC
relationships; hence, shapefiles A and C (detail above) included areas without SOC predictions
so the percentage of the marsh area for which SOC predictions were made was <100 %.
The SCSP shapefile was built by combining three GIS layers: i) the first layer provided the
distribution of saltmarsh areas in England and Wales, and is distributed by the Environmental
Agency (EA) (available at https://data.gov.uk/dataset/saltmarsh-extents1); ii) the second
layer gave the distribution of NVC classes in Welsh salt marshes, and was provided by Natural
Resources Wales ('Intertidal Phase-2' shapefile); and iii) the third layer provided simplified
soil type information, and was obtained from 'Soilscapes', a 1:250,000 scale, soil map covering
England and Wales, and developed by LandIS (http://www.landis.org.uk/). The EA shapefile
(i) represented saltmarsh areal extent as measured between 2006 and 2009 across England





and Wales (Phelan et al., 2011). The phase-2 survey data of NVC communities (ii) were derived
from 1996-2003 surveys of saltmarsh plant carried out for all of Wales (Brazier et al., 2007).
Soils of the Soilscape map (iii) were simplified into the two types used in SOC-predicting
algorithms: sandy or non-sandy soil. Comparison between mapped soil types and simplified
soil types measured in the field are shown in Table S1 (Supplement). The SCSP shapefile and
instructions on how to use it are available at https://www.saltmarshapp.com/saltmarsh-
tool/.

**3 Results**
3.1. Site characterisation
Plant and soil characteristics for each vegetation type of the 23 saltmarsh sites are shown in
Table S2, Supplement. SOC stock was often greater in both *J. gerardii* (SM16) and *J. maritimus*
(SM15; SM18) plant communities (40-60 t C ha$^{-1}$) than in the *Atriplex* (SM14) and *Puccinellia*
(SM13) communities (20-50 t C ha$^{-1}$). Soil pH of 6-7.5 was common throughout, but electrical
conductivity (a proxy for soil salinity) was more variable, dependent on specific position and
elevation relative to the tidal frame. Plant species richness was consistent across *P. maritima*,
*J. gerardii* and *J. maritimus* communities (4 – 10 species m$^{-2}$) with only *A. portulacoides*
occurring commonly as a monoculture. Plant height was variable, between 3-30 cm for *P.*
*maritima* and *J. gerardii*, with shorter swards when grazers present. *A. portulacoides* shrubs
were consistently 20-30 cm high, with *J. maritimus* tussocks 40-70 cm tall. Root biomass of
between 1-5 kg DW m$^{-2}$ was common, with *J. gerardii* and *J. maritimus* communities typically
having greater root biomass than the other two community types.



### 3.2. SOC stock: explanatory variables and model predictions


The relationship between the response variable 'SOC stock' and the plant and soil explanatory
variables was quantified by 6 uni- and 4 bi-variate models (Table 3). Assessment of 'vegetation
type' (Veg_model) or 'NVC class' (NVC_model) alone accounted for 36-37 % of the variation
in SOC stock. Root biomass alone (Root_model) explained 32 % of variation. Simplified soil
type alone (Soil_model), where soil was divided into sandy or non-sandy groups, explained 32
% of variation rising to 45 % when texture categories (Text_model) were considered. Plant
species richness alone (Species_model) explained 41 % of variation in SOC stock (Fig. S2,
Supplement). Bivariate models including plant community variables (vegetation type or NVC
class) and simplified soil type (Veg_soil_model and NVC_soil_model) explained 40-44 % of
SOC stock, rising to 51-52 % when plant variables were coupled with soil texture category
(Veg_text_model and NVC_text_model).

### 3.3. Prediction of SOC stock: the SCSP tool and Saltmarsh App


The SCSP tool look up table (Table 4) provides a straightforward way to determine SOC stock
in a UK saltmarsh based on information on either simplified soil type, plant community (NVC
class or vegetation type) or both. For convenience the SCSP look up table also contains the
model used in the carbon calculator component of The Saltmarsh App (Veg_soil_model).
Predictions of SOC stock based on plant NVC communities (5 classes) produced SOC stock
predictions (top 10 cm of soil) varying from 32 t C ha$^{-1}$ for the *A. portulacoides* NVC class to
50 t C ha$^{-1}$ for the *J. gerardii* NVC class (Table 4). Predictions based on simplified soil types (2
types) predicted that sandy soils store less SOC (29 t C ha$^{-1}$) than non-sandy soils (43 t C ha$^{-1}$).
A series of GIS based maps, illustrating SOC stock (t C ha$^{-1}$) and total SOC stored per marsh (t
C) for all Welsh saltmarshes (based on three models: NVC_model; Soil_model;





NVC_soil_model) can be viewed in the Supplement, Fig. S3-S25 inclusive (exemplar Fig. 2) or
online at https://www.saltmarshapp.com/saltmarsh-tool/
**Table 3.** Six explanatory variables of SOC stock (t C ha$^{-1}$; top 10 cm of soil) in Welsh
saltmarshes, based on ANOVA output from mixed effect models, with F statistic values
presented.

| Model name | Vegetation type | NVC class | Plant species richness m$^2$ | Root biomass (kg DW m$^{-2}$) | Simplified soil type | Soil texture category | R$^2$ |
|---|---|---|---|---|---|---|---|
| *SOC stock prediction: 6 single variable models* | | | | | | | |
| Veg_model | 9.33 *** | | - | - | - | - | 0.36 |
| NVC_model | - | 7.84 *** | - | - | - | - | 0.37 |
| Species_model | - | | 9.61 ** | - | - | - | 0.41 |
| Root_model | - | | - | 15.0 *** | - | - | 0.32 |
| Soil_model | - | | - | - | 12.52 *** | - | 0.32 |
| Text_model | - | | - | - | - | 2.90 ** | 0.45 |
| *SOC stock prediction: 4 bivariate models* | | | | | | | |
| Veg_soil_model | 10.18 *** | | - | - | 22.39 *** | - | 0.40 |
| Veg_text_model | 10.66 *** | | - | - | - | 3.84 *** | 0.51 |
| NVC_soil_model | - | 9.17 *** | - | - | 22.54 *** | - | 0.44 |
| NVC_text_model | - | 7.92 *** | - | - | - | 3.63 *** | 0.52 |

Significance (** = p <0.01, *** = p <0.001)
Vegetation type (4 levels: *P. maritima*; *A. portulacoides*; *J. maritimus*; *J. gerardii*)
NVC class (5 levels: SM13; SM14; SM15; SM16; SM18)
Simplified soil type (2 levels: 'Sandy' soil with ≥0.45 sand; 'Non-sandy' soils with <0.45 sand including loam,
clay, organic soils)
Soil texture category (12 levels: *see Table 2*)





**Table 4.** SCSP tool look up table based on models of SOC stock prediction in Welsh salt
marshes (using output of a sub-set of models from Table 3).

| Vegetation type | NVC class | Simplified soil type | Model Coefficient(s) | | Model Intercept | Predicted SOC stock (t C ha$^{-1}$) |
|---|---|---|---|---|---|---|
| NVC_model: 'NVC class' only [p < 0.001, r$^2$ = 0.37, mean model standard error (SM13 ± 2.9, SM14 ± 3.9, SM15 ± 4.9, SM18 ± 3.4, SM16 ± 3.2)] | | | | | | |
| - (*P. maritima*) | SM13 | - | - | - | 39.5 | 40 |
| - (*A. portulacoides*) | SM14 | - | - | -7.8 | 39.5 | 32 |
| - (*J. maritimus*) | SM15 | - | - | -2.3 | 39.5 | 37 |
| - (*J. maritimus*) | SM18 | - | - | 9.3 | 39.5 | 49 |
| - (*J. gerardii*) | SM16 | - | - | 10.4 | 39.5 | 50 |
| Soil_model: 'Simplified soil type' only [p < 0.001, r$^2$ = 0.32, mean model standard error ± 3.9] | | | | | | |
| - | - | Sandy | - | - | 29.4 | 29 |
| - | - | Non-sandy | - | 13.7 | 29.4 | 43 |
| Veg_soil_model: 'Vegetation type' and 'Simplified soil type' [p < 0.001, r$^2$ = 0.4, mean model standard error (*P. maritima* ± 2.7, *A. portulacoides* ± 3.3, *J. maritimus* ± 3.3 , *J. gerardii* ± 3.0)] | | | | | | |
| *P. maritima* | - (SM13) | Sandy | 8 | -12.9 | 32.7 | 28 |
| *P. maritima* | - (SM13) | Non-sandy | 8 | 12.9 | 19.8 | 41 |
| *A. portulacoides* | - (SM14) | Sandy | - | -12.9 | 32.7 | 20 |
| *A. portulacoides* | - (SM14) | Non-sandy | - | 12.9 | 19.8 | 33 |
| *J. maritimus* | - (SM15 & SM18) | Sandy | 15.1 | -12.9 | 32.7 | 35 |
| *J. maritimus* | - (SM15 & SM18) | Non-sandy | 15.1 | 12.9 | 19.8 | 48 |
| *J. gerardii* | - (SM16) | Sandy | 16.3 | -12.9 | 32.7 | 36 |
| *J. gerardii* | - (SM16) | Non-sandy | 16.3 | 12.9 | 19.8 | 49 |
| NVC_soil_model: 'NVC class' and 'Simplified soil type' [p < 0.001, r2 = 0.44, mean model standard error (SM13 ± 3.3, SM14 ± 3.7, SM15 ± 5.2, SM18 ± 3.3, SM16 ± 3.4)] | | | | | | |
| - (*P. maritima*) | SM13 | Sandy | - | -14.1 | 40.4 | 26 |



| - (*P. maritima*) | SM13 | Non-sandy | - | 14.1 | 26.3 | 40 |
|---|---|---|---|---|---|---|
| - (*A. portulacoides*) | SM14 | Sandy | -7.2 | -14.1 | 40.4 | 19 |
| - (*A. portulacoides*) | SM14 | Non-sandy | -7.2 | 14.1 | 26.3 | 33 |
| - (*J. maritimus*) | SM15 | Sandy | 2.4 | -14.1 | 40.4 | 29 |
| - (*J. maritimus*) | SM18 | Sandy | 10.1 | -14.1 | 40.4 | 36 |
| - (*J. maritimus*) | SM15 | Non-sandy | 2.4 | 14.1 | 26.3 | 43 |
| - (*J. maritimus*) | SM18 | Non-sandy | 10.1 | 14.1 | 26.3 | 50 |
| - (*J. gerardii*) | SM16 | Sandy | 9.5 | -14.1 | 40.4 | 36 |
| - (*J. gerardii*) | SM16 | Non-sandy | 14.1 | 9.5 | 26.3 | 50 |

Variables not in model denoted by '-'; Variables related to 'Vegetation type' or 'NVC class' but not included in
analysis in parentheses '()'.
Vegetation type (4 levels: *P. maritima*; *A. portulacoides*; *J. maritimus*; *J. gerardii*)
NVC class (5 levels: SM13; SM14; SM15; SM16; SM18)
Simplified soil type (2 levels: 'Sandy' soil with ≥0.45 sand; 'Non-sandy' soils with <0.45 sand including loam,
clay, organic soils)




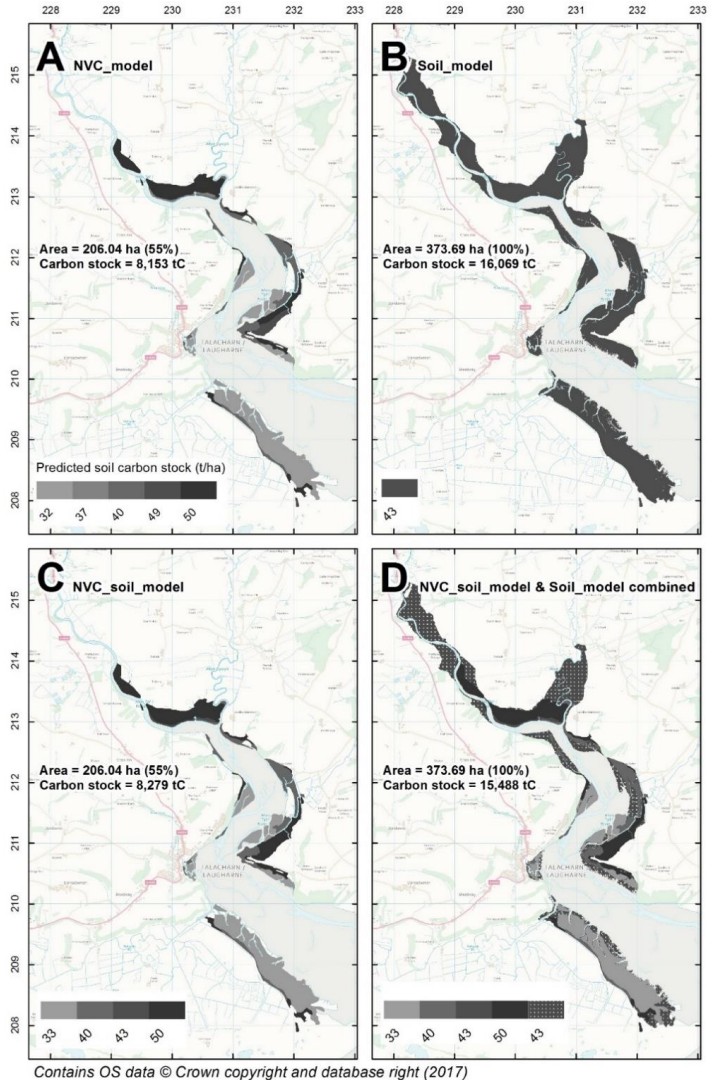

Contains OS data © Crown copyright and database right (2017)

**Figure 2.** Predictions of SOC stock (t C ha$^{-1}$ for top 10 cm) for saltmarshes at Laugharne in

south Wales. SOC stock was predicted by **A**) 'NVC class' only (NVC_model); **B**) 'Simplified soil

type' only (Soil_model); **C**) 'NVC and simplified soil type' combined, (NVC_soil_model); **D**)

'NVC and simplified soil type' combined (NVC_soil_model) plus predictions based on

'simplified soil type' (Soil_model) where SOC predictions for NVC pioneer communities were

not known. Inserted into maps are estimates of the total amount of SOC (t C) for all marshes





visible, for the 'Area' of the saltmarsh (%) for which we had the necessary information to
make predictions. Laugharne marsh included NVC communities for which the study did not
have predictive SOC to NVC relationships; hence, panel A and C include areas without SOC
predictions (white colour) and the percentage of the marsh area for which SOC predictions
were made are <100 %.

**4 Discussion**
The accurate prediction of 'blue carbon' stock is of interest to a wide range of stakeholders
including the IPCC (2014). This study has demonstrated that a large proportion of the variation
in SOC stock in saltmarsh habitats can be predicted from just two easy-to-measure variables,
plant community ('vegetation type' or 'NVC class') and simplified soil type, which together
accounted for close to half of the variation in SOC stock in 23 Welsh salt marshes. Associations
of SOC with plant and soil characteristics have been demonstrated in other ecosystems
(Amundson, 2001; Bai et al., 2016; Manning et al., 2015), although this study is the first to use
such relationships to produce a national inventory of blue carbon storage, with previous
attempts largely unsuccessful (Serrano et al. 2016).
4.1. Ecological observations
Whilst SOC stock in UK saltmarshes was broadly predicted by soil type, with non-sandy soils
more carbon rich, there remained a clear association between SOC stock and plant
community type, with rush-dominated *J. maritimus* and *J. gerardii* communities associated
with greater SOC stocks than either *A. portulacoides* or *P. maritima* communities. The deep-
rooted saltmarsh shrub *A. portulacoides* (Decuyper et al., 2014) occurred predominantly as a



near monoculture (Ford et al., 2016), with the shallow-rooted salt marsh grass *P. maritima*
community found alongside simple-rooted plants such as *Plantago maritima*. In contrast, the
rushes *J. gerardii* and *J. maritimus*, characterised by extensive laterally creeping rhizomes with
thick anchors and many shallow fine roots, commonly grew alongside the grasses *Festuca*
*rubra* and *Agrostis stolonifera* and various other forbs. The diverse *Juncus* communities are
known to have a wide variety of rooting strategies (Minden et al., 2012) that lead to greater
root biomass and consequently greater SOC stock (Jones and Donnelly, 2004; Loreau et al.,
2001). Higher SOC stock in *Juncus* areas might also arise as these species grow in waterlogged
conditions that limit aerobic breakdown of organic material (Ford et al., 2012), while *A.*
*portulacoides* is known to colonise relatively well-aerated and drained areas (Armstrong et
al., 1985).
4.2. Tools for broad-scale predictions of saltmarsh SOC stock
The study findings were used to develop two practical tools for predicting the SOC stocks of
salt marshes: the SCSP tool for expert stakeholders (i.e. IPCC, blue carbon initiatives,
academics, policy makers and land managers), and the Saltmarsh App for the general public
(find both at https://www.saltmarshapp.com). All of the univariate and bivariate models
tested in this study explained ≥32 % of the variation in saltmarsh SOC stocks, however not all
were of practical use for the tool/app, which required variables that were either easy to
measure or readily available as GIS layers. For example, the characterisation of soils into 12
soil texture categories produced consistently better univariate and bivariate predictions of
SOC (~50% of variation explained) than simple classification into sandy or non-sandy soils
(~33%), as texture-classification allowed a more accurate assessment of the clay to sand ratio,
a key indicator of SOC (Arrouays et al., 2006; O'Brien et al., 2015). However, the 2-class



simplified soil type classification was selected for use in the tools, as existing UK soil maps
categorised saltmarsh soils in these terms, and because non-specialists can distinguish sandy
from non-sandy soils in the field. For plant community type, predictions by 'vegetation type'
or 'NVC class' performed equally well, both explaining over a third of variation in SOC in
univariate models, rising to nearly half when combined with either simplified soil type or
texture classification. NVC class was selected as a key variable for SCSP as it is often mapped
at UK level by national agencies, whereas the easier to identify vegetation type was chosen
for the Saltmarsh App. In summary, the SCSP tool generates predictions and maps of
saltmarsh SOC stock from existing mapped information on soil type, NVC classification, or
both. The Saltmarsh App predicts SOC stock from field-based information on vegetation type
and simplified soil type combined.
4.2. Advantages and limitations of predicting blue carbon from vegetation and soil types
Coastal vegetated habitats are now increasingly acknowledged as important carbon sinks
(Howard et al., 2017), based on their high primary production, sediment trapping capacity
and the biogeochemical conditions of their sediments, which slow the decay of organic
material (Kelleway et al., 2017, McLeod et al., 2011). The contribution of coastal habitats,
such as salt marshes, to climate change mitigation had previously been under-estimated
(Scholefield et al., 2013), mainly due to their relatively small area cover relative to the open-
ocean or terrestrial vegetated ecosystems. However, on a per area basis, coastal wetlands
equate to similar or more efficient carbon sinks than most terrestrial forests (Mcleod et al.,
2011; Pan et al., 2011). Indeed, this study shows Welsh marshes hold up to 50 t C ha$^{-1}$ in the
top 10 cm of soil, equivalent to carbon densities in habitats such as fresh-water wetlands,
semi-natural grasslands and woodlands (Ostle et al., 2009). The SOC predictive models and



associated tool presented in this paper are widely applicable to other UK salt marshes, but
also throughout north-western European salt marshes (from Portugal to the Baltic), due to
the similarity of common and wide-spread vegetation types (Adam, 1990). However, for use
in other biogeographical regions, particularly North America, where salt marshes are
dominated by large *Spartina* species that produce organogenic soils (Adam, 1990), the
methods would need further ground-truthing.
The SCSP tool provides SOC predictions for saltmarsh plant communities indicative of the low,
mid and high marsh zones, representing around half of the total Welsh saltmarsh area (Brazier
et al., 2007). However, future work could boost the scope of the SCSP by validating SOC stock
predictions for pioneer communities (*Spartina* and *Salicornia*), that may differ markedly in
biotic indicators of SOC stock such as root biomass (Keiffer and Ungar, 2002; Schwarz et al.,
2015). At present, pioneer communities are defined by simplified soil type alone (see panel D
in Fig. 2). Common to many ecosystem service mapping tools, the SCSP tool assumes linearity
of the relationship between area and ecosystem service, this however is uncertain (Barbier et
al., 2008; Koch et al., 2009), and should be the next frontier of ecosystem service research.
While the SCSP tool has advantages in terms of translating ecology into practitioner-ready
information, something that is increasingly being demanded of ecologists (see Chapin, 2017,
and the Special Issue on 'translational ecology' in Frontiers in Ecology and Environment,
December 2017), such an approach also has some limitations. Namely, in the process of
translating ground level observations of ecosystem benefits (e.g. SOC stocks) into large-scale
maps, there is some information that gets 'lost in translation' (*sensu* Jackson et al., 2017). In
the case of this study, we were inherently limited by the need to use a reduced number of
the simplest variables available to any practitioner (e.g. vegetation community type), and at



the same time, variables that feature in national cartographic programmes (e.g. coarse soil
categories maps). Even so, the simple models selected for the SCSP tool explained ~50% of
the variation in SOM in the studied salt marshes. However, there is still another 50% that we
do not account for in this work. We know some of this variation is explained by the need to
use simplified soil categories (instead of soil texture) and the inability to use root biomass and
plant species richness as variables in the final tool (as these variables need more expertise to
estimate, and do not feature in an available GIS layer). The rest of the variation in SOC stock
might be attributed to differences in land use (i.e. grazed vs. un-grazed marshes) (Davidson
et al., 2017; Mueller et al., 2017), differences in marsh elevation within the tidal frame, or in
the geomorphological context of the marsh (e.g. fringing or estuarine, and if estuarine, near
the mouth of the estuary or towards the head of the estuary) (Arriola and Cable, 2017),
salinity or pH (Chambers et al., 2013), level of urbanisation of the catchment (Deegan et al.,
2012), past history of the marsh (Kelleway et al., 2017), whether the marsh sits in a dynamic
or stable area (J.F. Pagès et al., unpublished manuscript), level of disturbance/exposure it is
being subjected to (Macredie et al., 2013), among other factors. Despite the caveats listed
above, this study has demonstrated the ability to predict up to half the variation in saltmarsh
SOC stock from very simple environmental metrics.

**5 Data availability**
The data are available by request from the corresponding author.
**The Supplement related to this article is available online.**



*Author contribution*. MS, AG and HF designed the experiment. HF, MD-E, JP and RH carried
out the experiment. HF and CL analysed data and created GIS maps. HF prepared the
manuscript with contributions from all co-authors.
*Competing interests*. The authors declare that they have no conflict of interest.
*Acknowledgements.* This study presents data collected as part of the Coastal Biodiversity and
Ecosystem Service Sustainability project (CBESS: NE/J015644/1), part of the BESS programme,
a 6-year programme (2011–2017) funded by the Natural Environment Research Council
(Bangor University grant reference: NE/J015350/1) and the Biotechnology and Biological
Sciences Research Council (BBSRC) as part of the UK's Living with Environmental Change
(LWEC) programme. The views expressed are those of the authors and do not reflect the views
of BESS Directorate or NERC. The authors also acknowledge financial support from the Welsh
Government and Higher Education Funding Council for Wales through the Sêr Cymru National
Research Network for Low Carbon, Energy and Environment. Thanks also to the National
Trust, Natural Resources Wales, the Ministry of Defence, county councils, private estates and
farmers for access to their land.

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
