# Peer review of "Large-scale predictions of saltmarsh carbon stock based on simple observations of plant"

_Biogeosciences, 2018_

## Referee Comment (RC1) · Anonymous Referee #1 · 14 Sep 2018

General comments:

Blue C ecosystems show higher rates of C sequestration than many other ecosystems on the long-term. That is, these systems build up with rising sea level, their soils do not become C saturated (as terrestrial soils) and thus C sequestration in soils can be maintained over centuries or millennia. A central driver of C sequestration in these systems is therefore accretion, which I do not see considered in this manuscript. This needs consideration in the discussion part.

A related point is that only top soil (10 cm) C contents were assessed, so only a small fraction. "Carbon stock" is therefore misleading, particularly regarding the often several

meters deep soils of high C density typical for blue C ecosystems. Also, several studies demonstrated sharp declines in C density/content with soil depth in tidal wetlands, so that little information about the total C stock can be inferred from the top-soil C content. The focus on top soils needs to be made clear from the beginning of the ms throughout, and the implications of the strongly limited data set (i.e. missing depth assessment) need to be discussed. The relevance despite this limitation needs to be demonstrated.

I am not sure if the application of the results (i.e the SCSP and the Salt Marsh App) are in the scope of this journal. These two parts of the work might be more appropriate for a Methods journal, but I leave this decision to the Editor.

Specific points:

42: add pioneer works, i.e. Chmura et al. 2003

77-80: So how deep do these plants root in relation to your sampling depth of 10 cm? How much of the belowground biomass stock can be captured?

101-103: check sentence

132: Craft et al 1991 (Loss on ignition and kjeldahl digestion for estimating organic carbon and total nitrogen in estuarine marsh soils: Calibration with dry combustion) demonstrate that the SOM-SOC relationship depends on soil type and that the use of a simple conversion factor (i.e. 55%) can lead to both strong under- and overestimation of SOC. This needs consideration.

161: This is R code not English. Please make it understandable for people using other software throughout your methods section.

178: It is unclear what is new about the SCSP tool in this manuscript that has not been described in Skov 2016?

308: deep-rooting would lead to a C allocation in the soil profile that you did not capture. It is possible that the C stocks under your different plant communities are not different

but you could not capture this with your sampling design

352: please be more specific: long-term C sequestration (aka C burial) is higher than in forests.

356-58: You did not demonstrate that your data are applicable to other UK marshes outside Wales or even European marshes in general.

360: Spartina is also a dominant genus in many European marshes.

―――――――――――――――――――

---

## Referee Comment (RC2) · Anonymous Referee #2 · 30 Sep 2018

Major comments This is an important topic and it would be useful to have good predictors for a region, however, the information provided does not yet support the validity of a national inventory.

Methods There is no mention of how the locations of transects and quadrats were chosen. Methods suggest that vegetation types were specifically chosen, but later (ln 118) it is mentioned that an analysis was conducted to determine how they fit in NVC classes. This sounds a bit circular. Were vegetation types specifically targeted? It is not clear why the statistical analyses had to be restricted to a linear model (ln 150) – it should not be restricted to because citizen scientists might use it- application of

models is not commonly tasks that citizen scientists perform. If so, authors could provide a spreadsheet to perform the calculation. Location was divided into two classes, north and south Wales, and entered as a categorical variable. Is there a major biogeographical change between north and south? If latitude was considered important why not simply use latitude, rather than using a categorical value, to increase the ability to distinguish a gradient?

Vegetation covered To determine how geographically broad the results of this study could be one, needs to know more about the vegetation sampled, and that not sampled. Only 5 salt marsh vegetation classes are listed in this study – all simply identified by a dominant(?) species – two are identified by the same species, Juncus maritimus. How many quadrats were sampled in each vegetation class and were these equally distributed among the marshes? What proportion of cover is attributed to the dominant species? What types of species occur with the dominant? It would be useful to provide a table showing typical species composition and cover. It is likely that perennials will contribute more to soil carbon than annuals and graminoids over forbs (although Triglochin and Plantago can have substantial belowground biomass). Species richness was not found to be a significant explanatory variable, but what about the proportion of perennial vs annual plants? How many NVCs are there in UK salt marshes?

Breadth of Geographical Application Authors suggest that their model can be used to estimate carbon stocks in the UK and perhaps northwestern Europe, as well. Yet, not all plant communities present in Welsh salt marshes were sampled (ln 289). What communities does this study miss from Wales and across the UK? How much salt marsh area is not accounted for? Authors further that their model could be applied from the Baltic to Portugal – is the vegetation really that consistent?

Unexplained variability Authors seem to have preliminarily truncated statistical analyses for this study. They note in the Discussion that ∼50% of the variation in the marshes they studied has yet to be statistically explained, further noting that the rest of the variation might be attributed to differences in grazing, salinity, pH, geomorphological context, level of urbanisation, past disturbance, whether in a dynamic or stable area. Authors have reported data on grazing, salinity and pH that could easily be assessed in an expanded model. Geomorphological context can easily be determined from the maps in the supplementary material. As they mention "level of urbanisation" in the context of the study by Deegan et al. (2012). I assume they refer to nutrient loading of the estuary. Nutrient loading is not limited to urban development, but also to agricultural uses. If watershed nutrient loading models have been developed for UK estuaries the nutrient loading could be assessed as a predictor as well. Level of disturbance/exposure seems to be similar to "whether the marsh sits in a dynamic or stable area", something that could be determined fairly easily.

Soil Carbon IPCC guidelines for calculation of greenhouse gas emission from land use change in coastal wetlands (Kennedy et al. 2013) suggest stocks be considered over 1 m depth. Granted such depths are difficult to sample and accurately measure bulk density, but not all soil samples in this study reached even 10 cm depth, yet this study is supposedly focussed on the upper 10 cm of soil. And, different soil parameters were measured over different depths. It is not clear how soil was sampled to determine bulk density over 10 cm depth – and this is a very critical element, central to the entire study. Text states that soil was collected from 2 cm to 9.5 cm. I suspect that the sampling ring mentioned was not 3.1 cm high but 7.5 cm high (diameter and height reversed in text?). Soil organic carbon was determined from this sample, as well. This is not quite 10 cm and why was the surface 2 cm not collected? The bulk density and soil carbon measurements do not correspond to the soil texture which was determined only on the surface 5 cm (ln 133). Do authors have any idea what the soil is like below 10 cm depth? Are any of the sampled marshes filled or previously drained ad now restored? Did Emmett et al. (2010) establish a relationship between OC and LOI to derive the conversion of 55% (Ln 131)?

First National Inventory of blue carbon storage? It is a bit preliminary for authors to claim to have the first national inventory of blue carbon storage.

Technical Editing Figure 1 fig b needs a scale bar Figure 2 compares carbon stocks at a single marsh applying results of different models. However, because the areas covered are different it is not a fair comparison of the difference in carbon stocks predicted by the model. Ln 41 Soil organic carbon IS belowground Ln 45 I am surprised that salt marshes are considered terrestrial habitats Ln 87 What current inventory? Ln 121 samples are dried to there is no longer a loss of moisture rather than for a prescribed time –Did authors assess whether 72 hrs adequate? Ln 367 what is meant by a "pioneer community" here? Ln 390 Since level of disturbance/exposure seems to be similar to "whether the marsh sits in a dynamic or stable area" seem to be the same there is no reason to cite an unpublished manuscript. Ln 269 shouldn't 0.45 be 45%?

References cited Kennedy HA, Alongi DM, Karim A, Chen G, Chmura GL, Crooks S, Kairo JG, Liao B, Lin G. 2013. Chapter 4 Coastal Wetlands In: Supplement to the 2006 IPCC Guidelines on National Greenhouse Gas Inventories: Wetlands.

---

## Author Comment (AC1) · 19 Oct 2018

Anonymous Referee #1 General comments:

Blue C ecosystems show higher rates of C sequestration than many other ecosystems on the long-term. That is, these systems build up with rising sea level, their soils do not become C saturated (as terrestrial soils) and thus C sequestration in soils can be maintained over centuries or millennia. A central driver of C sequestration in these systems is therefore accretion, which I do not see considered in this manuscript. This needs consideration in the discussion part.

[Figure]

Thank you for your comment. We will add a short section in the Discussion on the importance of accretion, although the key focus of this paper remains the prediction of soil organic carbon (SOC) stock from simple observations of plant community and soil type.

A related point is that only top soil (10 cm) C contents were assessed, so only a small fraction. "Carbon stock" is therefore misleading, particularly regarding the often several meters deep soils of high C density typical for blue C ecosystems. Also, several studies demonstrated sharp declines in C density/content with soil depth in tidal wetlands, so that little information about the total C stock can be inferred from the top-soil C content. The focus on top soils needs to be made clear from the beginning of the ms throughout, and the implications of the strongly limited data set (i.e. missing depth assessment) need to be discussed. The relevance despite this limitation needs to be demonstrated.

We thank the reviewer for this comment and acknowledge that we need to make this and the following associated points clearer in the text. The blue carbon literature shows SOC stock in the top layer of soil is generally indicative of SOC stock in deeper soil layers; SOC typically has a long-linear relationship with depth. We will provide evidence of literature demonstrating this principle and will add a section in the Introduction and Discussion to elaborate on this point. We will substantiate the principle further by providing examples of SOC to soil depth profiles from the study sites involved (data source: Kingham 2013 The broad-scale impacts of livestock grazing on saltmarsh carbon stocks. PhD thesis, Bangor University, UK). Kingham (2013) sampled 224 cores from 22 saltmarshes in the study region, differing in soil type, plant community type and grazing intensity. By reanalysing this data set, we will show that sampling to a depth of 10 cm consistently captures $72 \pm 1$ % of total soil organic carbon (Figure S1). We thus argue that surface SOC stock can provide a reliable predictor of deeper carbon stores and is therefore a useful indicator of total SOC stock for UK saltmarshes.

I am not sure if the application of the results (i.e the SCSP and the Salt Marsh App) are in the scope of this journal. These two parts of the work might be more appropriate for
a Methods journal, but I leave this decision to the Editor.

We believe that the manuscript is strengthened by the inclusion of reference to the Saltmarsh Carbon Stock Predictor (SCSP) and the Salt Marsh App. These tools are practical applications of the principles demonstrated in the paper. They give opportunity for large-scale prediction of blue carbon stores by non-experts either in the field or from existing maps. Reference to SCSP and the App should make the paper more attractive to land managers.

Specific points: 42: add pioneer works, i.e. Chmura et al. 2003

We will add 'Chmura et al. (2003) Global carbon sequestration in tidal, saline wetland soils' to the reference list and cite it in line 42.

77-80: So how deep do these plants root in relation to your sampling depth of 10 cm? How much of the belowground biomass stock can be captured?

Sampling to 10 cm consistently incorporates $60 \pm 1.5$ % of total root biomass (Kingham, 2013), this is detailed in a later comment relating to root biomass.

101-103: check sentence

The phrase 'quadrat-scale' is not clearly defined here. We will improve the wording if asked to edit the submitted manuscript by the editor.

132: Craft et al 1991 (Loss on ignition and kjeldahl digestion for estimating organic carbon and total nitrogen in estuarine marsh soils: Calibration with dry combustion) demonstrate that the SOM-SOC relationship depends on soil type and that the use of a simple conversion factor (i.e. 55%) can lead to both strong under- and overestimation of SOC. This needs consideration.

We would argue that for our quick-and-easy estimation of soil carbon stock from vegetation and simplified soil type a simple conversion factor (between soil organic matter % and soil C %) is adequate. Variation is apparent in the literature on conversion rates

between soil organic matter (calculated from LOI) and soil C (%) in saltmarsh habitats (Coastal Blue Carbon methods for assessing carbon stocks and emissions factors in mangroves, tidal salt marshes, and seagrass meadows, the Blue Carbon initiative https://www.cifor.org/library/5095/) with the majority of publications from the US, where soil types and dominant vegetation species are often different. The advantage of the 0.55 conversion is that it is from a UK source.

161: This is R code not English. Please make it understandable for people using other software throughout your methods section.

Apologies, I will re-write this section to make it clearer for people using other software whilst still referencing key R packages as requested by the R community (all software is open-source and created by experts free of charge so citations are requested etiquette). If editor requests it this detail can be moved to a supplement.

178: It is unclear what is new about the SCSP tool in this manuscript that has not been described in Skov 2016?

Skov et al. (2016) is a user manual for the SCSP. While it does give a simple overview of the underpinning principles and principles of the analysis for the tool, it does not go into any depth about the analytical components, nor does it introduce or discuss in any depth the underpinning literature.

308: deep-rooting would lead to a C allocation in the soil profile that you did not capture. It is possible that the C stocks under your different plant communities are not different but you could not capture this with your sampling design Kingham (2013; full reference above) analysis of 224 cores from 22 marshes within the study regions showed that sampling to a depth of 10 cm captured 60 $\pm$ 1.5 % of root biomass (Figure S2). This finding can be included in the manuscript and detailed in an appendix. Shallow root biomass is therefore broadly indicative of deeper root biomass, allowing us to assess differences in root biomass between plant communities using relatively shallow cores.

352: please be more specific: long-term C sequestration (aka C burial) is higher than in forests.

The current statement 'However, on a per area basis, coastal wetlands equate to similar or more efficient carbon sinks than most terrestrial forests (Mcleod et al., 2011; Pan et al., 2011)' could be altered to read: 'However, on a per area basis, coastal wetlands are more efficient carbon sinks than most terrestrial forests (Mcleod et al., 2011; Pan et al., 2011) due to their ability to accrete vertically in response to sea level rise (Chmura et al. 2003)'.

356-58: You did not demonstrate that your data are applicable to other UK marshes outside Wales or even European marshes in general.

This study demonstrates the principle that carbon is predictable from vegetation and soil types common across the UK and North-west Europe. We have data to present within the supplement to show comparability between Wales and saltmarshes in other UK regions (this was not included in the present version), see Figure S3. All estimates are close apart from the 'P. maritima + Sandy soil' category, possibly indicative of heavy livestock-grazing on the marsh where this combination was most common.

360: Spartina is also a dominant genus in many European marshes. We agree with the reviewer. We have been up-front in the manuscript about the lack of representation of European pioneer plant communities including Spartina anglica. We will alter the text at line 360 to reflect this. However, in this instance, we were referring to the fact that American marshes, tend to be dominated by Spartina species that render the soil organogenic, more than minerogenic. This has been shown to impact carbon burial (Davidson et al 2017), and we just wanted to make the reader aware of the different functioning of European and American marshes that might mean our method has to be adapted there.

[Figure]

[Figure]

[Figure]

**Fig. 1.** Figure S1. Soil carbon (%) accounted for by soil depth based on 224 saltmarsh cores from 22 saltmarshes in the study region, differing in soil type, plant community type and grazing intensity.

[Figure]

**Fig. 2.** Figure S2. Root biomass (%) accounted for by soil depth based on 224 saltmarsh cores from 22 saltmarshes in the study region, differing in soil type, plant community type and grazing intensity.

[Figure]

**Fig. 3.** Figure S3. Field measurements of soil carbon stock (n = 132) using data from three Lancashire and three Essex (UK) saltmarshes compared to predictions from the SCSP tool (Veg_soil_model – 'Vegetation

---

## Author Comment (AC2) · 19 Oct 2018

Major comments This is an important topic and it would be useful to have good predictors for a region, however, the information provided does not yet support the validity of a national inventory.

Thank you for your comments. We argue that the manuscript provides the basis for a national (Wales) inventory of blue carbon. We want to take the opportunity to highlight that the scale of our inventory is nearly unprecedented, and that the aim of our

study is to make available a very simple method to the community of managers and non-academic sectors, which might not be acquainted with (nor have the means to do) elemental analyses, core extraction, etc. but that do commonly have access to vegetation and soil maps. Our manuscript provides support to these simple to use methods that give an estimate of soil carbon stock where estimates were previously largely non-existent, particularly at an individual saltmarsh level.

Methods There is no mention of how the locations of transects and quadrats were chosen. Methods suggest that vegetation types were specifically chosen, but later (ln 118) it is mentioned that an analysis was conducted to determine how they fit in NVC classes. This sounds a bit circular. Were vegetation types specifically targeted?

The 4 vegetation types focused on in this study were located using governmental maps based on vegetation surveys from 1996-2003 (detailed in section 2.6). Vegetation type was therefore validated on the ground as species extent could have altered between the survey date and the present day.

It is not clear why the statistical analyses had to be restricted to a linear model (ln 150) – it should not be restricted to because citizen scientists might use it- application of models is not commonly tasks that citizen scientists perform. If so, authors could provide a spreadsheet to perform the calculation.

Mixed effects models were chosen as they allow the estimation of fixed and random effects on the response variable (in this case SOC). The use of linear models is widespread. They usually assume a linear relationship between x and y variables (although you can always use transformations to model non-linear relationships using the same techniques, where needed [not in this case]). In addition, the analysis of variance that typically follows model building allow presenting the results in a format that is readily interpretable by members of both academia and governmental organisations. We will improve the text to make clear that the only thing that we ask citizen scientists to do as part of the Saltmarsh App is to check soil texture and vegetation

type. Members of the public are not required to carry out their own analysis at all.

Location was divided into two classes, north and south Wales, and entered as a categorical variable. Is there a major biogeographical change between north and south? If latitude was considered important why not simply use latitude, rather than using a categorical value, to increase the ability to distinguish a gradient?

Location was included as part of the model structure with site nested within location (north or south). If location is removed P value category (i.e. P < 0.001, P < 0.01 etc.) and r2 values (to 2 significant figures) remain the same, we will therefore follow the reviewers suggestion and remove location from the model for the sake of simplicity.

Vegetation covered To determine how geographically broad the results of this study could be one, needs to know more about the vegetation sampled, and that not sampled. Only 5 salt marsh vegetation classes are listed in this study – all simply identified by a dominant(?) species – two are identified by the same species, Juncus maritimus. How many quadrats were sampled in each vegetation class and were these equally distributed among the marshes?

Part of this information is detailed in the supplementary material (Table S2). For the four vegetation types sampled, between 32 and 66 quadrats were surveyed, across a minimum of 9 and a maximum of 17 marshes, to reflect the dominant vegetation communities of the low, mid and high marsh saltmarsh communities along the coastline of Wales.

What proportion of cover is attributed to the dominant species? What types of species occur with the dominant? It would be useful to provide a table showing typical species composition and cover.

This information is detailed in Rodwell (2000), a table can be provided in supplementary material to determine each British National Vegetation Classification (NVC) with each subcategory. We can also use the maps in the supplementary material to provide an

estimate of the area coverage provided by the focus on 4 main vegetation types.

It is likely that perennials will contribute more to soil carbon than annuals and graminoids over forbs (although Triglochin and Plantago can have substantial below-ground biomass). Species richness was not found to be a significant explanatory variable, but what about the proportion of perennial vs annual plants? How many NVCs are there in UK salt marshes?

While from a purely academic point of view this is a relevant point, we should bear in mind the aim of making our methods applicable for managers and non-academics. Hence, given that the proportion of perennial to annual plants are mapped, the usefulness of this parameter usefulness in an applied tool such as the SCSP is limited. In contrast, NVCs are already mapped, what make them easy to use. There are 7 common NVCs found in the UK, 5 of which are considered in this manuscript. The two pioneer communities which were not assessed directly are included using mapped soil characteristics.

Breadth of Geographical Application Authors suggest that their model can be used to estimate carbon stocks in the UK and perhaps northwestern Europe, as well. Yet, not all plant communities present in Welsh salt marshes were sampled (ln 289). What communities does this study miss from Wales and across the UK? How much salt marsh area is not accounted for? Authors further that their model could be applied from the Baltic to Portugal – is the vegetation really that consistent?

Two pioneer saltmarsh communities common across Wales and the UK (Spartina and Salicornia), accounting for ∼30% area of Welsh saltmarsh, were not directly assessed in this study, however their soil carbon stock is predicted on the basis of mapped simplified-soil characteristics. The 5 common saltmarsh vegetation communities focused on in this study are also widespread across north-western Europe.

Unexplained variability Authors seem to have preliminarily truncated statistical analyses for this study. They note in the Discussion that _50% of the variation in the marshes

they studied has yet to be statistically explained, further noting that the rest of the variation might be attributed to differences in grazing, salinity, pH, geomorpho-logical context, level of urbanisation, past disturbance, whether in a dynamic or stable area. Authors have reported data on grazing, salinity and pH that could easily be assessed in an expanded model. Geomorphological context can easily be determined from the maps in the supplementary material. As they mention "level of urbanisation" in the context of the study by Deegan et al. (2012). I assume they refer to nutrient loading of the estuary. Nutrient loading is not limited to urban development, but also to agricul-tural uses. If watershed nutrient loading models have been developed for UK estuaries the nutrient loading could be assessed as a predictor as well. Level of disturbance/ exposure seems to be similar to "whether the marsh sits in a dynamic or stable area", something that could be determined fairly easily.

Wording in this section of the discussion does need to be improved as grazing, salinity and pH were indeed considered in early model selection but were not significant ex-planatory variables of soil carbon stock. Modelling geomorphological context, level of urbanisation and nutrient loading were considered beyond the scope of this manuscript which focuses on the prediction of soil carbon stock from simple measurements vege-tation type and simplified soil type.

Soil Carbon IPCC guidelines for calculation of greenhouse gas emission from land use change in coastal wetlands (Kennedy et al. 2013) suggest stocks be considered over 1 m depth. Granted such depths are difficult to sample and accurately measure bulk density, but not all soil samples in this study reached even 10 cm depth, yet this study is supposedly focussed on the upper 10 cm of soil. And, different soil parameters were measured over different depths.

We will add a section in the Introduction and Discussion to make clear that despite the fact that we are examining surface SOC stock, SOC stock in the top layer of soil is largely indicative of SOC stock in deeper soil layers. We will show that top SOC stock is indicative of deeper SOC stocks using data from Kingham, R.: The broadscale impacts of livestock grazing on saltmarsh carbon stocks. PhD thesis, Bangor University, UK, 2013. This thesis includes 224 samples from a range of UK saltmarshes differing in soil type, plant community type and grazing intensity. By reanalysing this data set, we show that sampling to a depth of 10 cm consistently captures 72 ± 1 % of total soil organic carbon (Figure S1). We thus argue that surface SOC stock can provide a reliable predictor of deeper carbon stores and is therefore a useful indicator of total SOC stock for UK saltmarshes.

It is not clear how soil was sampled to determine bulk density over 10 cm depth – and this is a very critical element, central to the entire study. Text states that soil was collected from 2 cm to 9.5 cm. I suspect that the sampling ring mentioned was not 3.1 cm high but 7.5 cm high (diameter and height reversed in text?). Soil organic carbon was determined from this sample, as well. This is not quite 10 cm and why was the surface 2 cm not collected?

If text is changed from 'vertically' to 'horizontally' this should improve the clarity of explanation. In addition we can add a diagram to the supplementary material. The bulk density core used was 3.1cm high and 7.5 cm in diameter, it was rotated into a horizontal position to quantify the top 2 – 9.5 cm of soil in line with the methods used in Ford et al. (2016) Soil stabilization linked to plant diversity and environmental context in coastal wetlands. Excluding the top 2cm ensured that above ground vegetation was not accidentally included within the bulk density core. The bulk density and soil carbon measurements do not correspond to the soil texture which was determined only on the surface 5 cm (ln 133). Do authors have any idea what the soil is like below 10 cm depth? Are any of the sampled marshes filled or previously drained ad now restored?

All of the saltmarshes in this study are semi-natural, i.e. they have not been filled or previously drained but are often managed in some way (grazing livestock for food production or conservation, right of way for coastal paths etc.). Soil texture was assessed by hand at ∼5cm depth to reflect the mid-point of the bulk density and soil carbon measurements (from a depth of 2cm to 9.5 cm depth to quantify the top 10cm of soil). This

method was used in Ford et al. (2016) Soil stabilization linked to plant diversity and environmental context in coastal wetlands.

Did Emmett et al. (2010) establish a relationship between OC and LOI to derive the conversion of 55% (Ln 131)?

LOI values were compared to total soil C content measured by elemental analyser in Emmett et al. (2010).

First National Inventory of blue carbon storage? It is a bit preliminary for authors to claim to have the first national inventory of blue carbon storage. We can change this to 'an inventory of blue carbon storage in surface soil layers'. We would like to emphasise that when we talk about National Inventory we do not mean UK-wide. We are talking of a National Inventory, for Wales (a nation within the UK). Technical Editing

Figure 1 fig b needs a scale bar.

A scale bar will be added to Figure 1b.

Figure 2 compares carbon stocks at a single marsh applying results of different models. However, because the areas covered are different it is not a fair comparison of the difference in carbon stocks predicted by the model.

Figure 2 illustrates soil carbon stock in two ways, firstly soil carbon stock is shown visually (in t C ha-1) using a grey scale with each section of marsh assigned a shade of grey based on predicted category of soil carbon stock, this scale is used regardless of saltmarsh area and matches the four models selected in this paper for use in the SCSP tool and the Saltmarsh App. Secondly each figure presents the area of the marsh in hectares alongside the total carbon stock (in the top 10 cm of soil) for that area in t C. Panel D illustrates 'best practice', where the NVC_soil_model is used where NVC communities are mapped, with Soil_model used for the remaining saltmarsh area when NVC information is not available (not mapped), thereby giving the best estimate of soil carbon stock for the whole marsh area.

Ln 41 Soil organic carbon IS belowground

This sentence will be altered to avoid confusion.

Ln 45 I am surprised that salt marshes are considered terrestrial habitats

In the Scholefield (2013) document they were classified as Coastal Margins alongside terrestrial habitats such as woodlands and grasslands.

Ln 87 What current inventory?

The wording in this sentence is a little misleading, it will be altered for clarity. We refer to the current inventory as information compiled by the IPCC (2014) on blue carbon storage.

Ln121 samples are dried to there is no longer a loss of moisture rather than for a prescribed time –Did authors assess whether 72 hrs adequate?

Drying at 60 or 70 °C for 72 hours is a commonly used methodology for drying plant vegetation. We always test the amount of time needed to get constant weight. In our experience, with the amount of plant material we use it rarely exceeds 72 hours. This was tested with a subset of samples at the beginning of this study and in this case, 72 hours was enough for the small amount of above-ground biomass analysed (<5 g fresh weight).

Ln 367 what is meant by a "pioneer community" here?

Pioneer communities are defined in the previous sentence as Spartina and Salicornia communities.

Ln 390 Since level of disturbance/exposure seems to be similar to "whether the marsh sits in a dynamic or stable area" seem to be the same there is no reason to cite an unpublished manuscript.

We will remove this reference from the manuscript.

Ln 269 shouldn't 0.45 be 45%?

Footnotes to tables will be edited to ensure use of either proportion (0.45) or percentage (45%) is clear.

References cited Kennedy HA, Alongi DM, Karim A, Chen G, Chmura GL, Crooks S, Kairo JG, Liao B, Lin G. 2013. Chapter 4 Coastal Wetlands In: Supplement to the 2006 IPCC Guidelines on National Greenhouse Gas Inventories: Wetlands. We will double check citation guidelines for this publication. At present we have cited it as IPCC (2014).

**Fig. 1.** Figure S1. Soil carbon (%) accounted for by soil depth based on 224 saltmarsh cores from 22 saltmarshes in the study region, differing in soil type, plant community type and grazing intensity.

---

## Author Response (AR1)

General comments:

Blue C ecosystems show higher rates of C sequestration than many other ecosystems on the long-term. That is, these systems build up with rising sea level, their soils do not become C saturated (as terrestrial soils) and thus C sequestration in soils can be maintained over centuries or millennia. A central driver of C sequestration in these systems is therefore accretion, which I do not see considered in this manuscript. This needs consideration in the discussion part.

*Thank you for your comment. We have added a sentence in the Discussion on the importance of accretion (line 374), although the key focus of this paper remains the prediction of soil organic carbon (SOC) stock from simple observations of plant community and soil type.*

A related point is that only top soil (10 cm) C contents were assessed, so only a small fraction. "Carbon stock" is therefore misleading, particularly regarding the often several meters deep soils of high C density typical for blue C ecosystems. Also, several studies demonstrated sharp declines in C density/content with soil depth in tidal wetlands, so that little information about the total C stock can be inferred from the top-soil C content. The focus on top soils needs to be made clear from the beginning of the ms throughout, and the implications of the strongly limited data set (i.e. missing depth assessment) need to be discussed. The relevance despite this limitation needs to be demonstrated.

*We thank the reviewer for this comment and have now made this and the following associated points clearer by referring to surface SOC stock (0-10 cm) in both the abstract (lines 17 and 22), the last paragraph of the Introduction (line 91) and throughout the Methods, Results and Discussion sections (highlighted in track changes). The blue carbon literature shows SOC stock in the top layer of soil is generally indicative of SOC stock in deeper soil layers; SOC typically has a long-linear relationship with depth. We have provided evidence of literature demonstrating this principle and have added a section to the Discussion (lines 384-395) to elaborate on this point. We substantiated the principle further by providing examples of SOC to soil depth profiles from the study sites involved (data source: Kingham 2013 The broad-scale impacts of livestock grazing on saltmarsh carbon stocks. PhD thesis, Bangor University, UK). Kingham (2013) sampled 224 cores from 22 saltmarshes in the study region, differing in soil type, plant community type and grazing intensity. By reanalysing this data set, we showed that sampling to a depth of 10 cm consistently captures 72 ± 1 % of total soil organic carbon (Figure S5), when sampling to a depth of 45 cm. We thus argue that surface SOC stock can provide a reliable predictor of deeper carbon stores and is therefore a useful indicator of total SOC stock for UK saltmarshes.*

I am not sure if the application of the results (i.e the SCSP and the Salt Marsh App) are in the scope of this journal. These two parts of the work might be more appropriate for a Methods journal, but I leave this decision to the Editor.

*We believe that the manuscript is strengthened by the inclusion of reference to the Saltmarsh Carbon Stock Predictor (SCSP) and the Salt Marsh App. These tools are practical applications*

*of the principles demonstrated in the paper. They give opportunity for large-scale prediction of blue carbon stores by non-experts either in the field or from existing maps. Reference to SCSP and the App should make the paper more attractive to land managers.*

Specific points:
42: add pioneer works, i.e. Chmura et al. 2003

*We have added 'Chmura et al. (2003) Global carbon sequestration in tidal, saline wetland soils' to the reference list and cite it in line 43.*

77-80: So how deep do these plants root in relation to your sampling depth of 10 cm? How much of the belowground biomass stock can be captured?

*Sampling to 10 cm consistently incorporates 60 ± 1.5 % of total root biomass (Kingham, 2013), this is detailed in a later comment relating to root biomass.*

101-103: check sentence

*The phrase 'quadrat-scale' is not clearly defined here. We have improve the wording to illustrate how measuring vegetation communities in the field allowed us to check the accuracy of existing NVC maps (see track changes lines 100-118).*

132: Craft et al 1991 (Loss on ignition and kjeldahl digestion for estimating organic carbon and total nitrogen in estuarine marsh soils: Calibration with dry combustion) demonstrate that the SOM-SOC relationship depends on soil type and that the use of a simple conversion factor (i.e. 55%) can lead to both strong under- and overestimation of SOC. This needs consideration.

*We would argue that for our quick-and-easy estimation of soil carbon stock from vegetation and simplified soil type a simple conversion factor (between soil organic matter % and soil C %) is adequate. Variation is apparent in the literature on conversion rates between soil organic matter (calculated from LOI) and soil C (%) in saltmarsh habitats (Coastal Blue Carbon methods for assessing carbon stocks and emissions factors in mangroves, tidal salt marshes, and seagrass meadows, the Blue Carbon initiative https://www.cifor.org/library/5095/) with the majority of publications from the US, where soil types and dominant vegetation species are often different. The advantage of the 0.55 conversion is that it is from a UK source.*

161: This is R code not English. Please make it understandable for people using other software throughout your methods section.

*Apologies, I have re-written this section to make it clearer for people using other software (lines 171-177) whilst still referencing key R packages as requested by the R community (all software is open-source and created by experts free of charge so citations are requested etiquette).*

178: It is unclear what is new about the SCSP tool in this manuscript that has not been described in Skov 2016?

*Skov et al. (2016) is a user manual for the SCSP. While it does give a simple overview of the underpinning principles and principles of the analysis for the tool, it does not go into any depth about the analytical components, nor does it introduce or discuss in any depth the underpinning literature.*

308: deep-rooting would lead to a C allocation in the soil profile that you did not capture. It is possible that the C stocks under your different plant communities are not different but you could not capture this with your sampling design

*Kingham (2013; full reference above) analysis of 224 cores from 22 marshes within the study regions showed that sampling to a depth of 10 cm captured 60 ± 1.5 % of root biomass (Figure S6), when sampled to 45 cm depth. This is now mentioned in the discussion (lines 392-393). Shallow root biomass is therefore broadly indicative of deeper root biomass, allowing us to assess differences in root biomass between plant communities using relatively shallow cores.*

352: please be more specific: long-term C sequestration (aka C burial) is higher than in forests.

*The current statement 'However, on a per area basis, coastal wetlands equate to similar or more efficient carbon sinks than most terrestrial forests (Mcleod et al., 2011; Pan et al., 2011)' has been altered to read: 'However, on a per area basis, coastal wetlands are more efficient carbon sinks than most terrestrial forests (Mcleod et al., 2011; Pan et al., 2011) due to their ability to accrete vertically in response to sea level rise (Chmura et al. 2003)' lines 372-374.*

356-58: You did not demonstrate that your data are applicable to other UK marshes outside Wales or even European marshes in general.

*This study demonstrates the principle that carbon is predictable from vegetation and soil types common across the UK and North-west Europe. We presented data within the supplement to show comparability between Wales and saltmarshes in other UK regions (this was not included in the previous version), see Figure S4 (line 378). All estimates are close apart from the 'P. maritima + Sandy soil' category, possibly indicative of heavy livestock-grazing on the marsh where this combination was most common.*

360: Spartina is also a dominant genus in many European marshes.

*We agree with the reviewer. We have been up-front in the manuscript about the lack of representation of European pioneer plant communities including Spartina anglica. We have altered the text at line 401 to reflect this. However, in this instance, we were referring to the fact that American marshes, tend to be dominated by Spartina species that render the soil organogenic, more than minerogenic. This has been shown to impact carbon burial (Davidson et al 2017), and we just wanted to make the reader aware of the different functioning of European and American marshes that might mean our method has to be adapted there.*

**Anonymous Referee #2**

Major comments This is an important topic and it would be useful to have good predictors for a region, however, the information provided does not yet support the validity of a national inventory.

*Thank you for your comments. We argue that the manuscript provides the basis for a national (Wales) inventory of blue carbon. We want to take the opportunity to highlight that the scale of our inventory is nearly unprecedented, and that the aim of our study is to make available a very simple method to the community of managers and non-academic sectors, which might not be acquainted with (nor have the means to do) elemental analyses, core extraction, etc. but that do commonly have access to vegetation and soil maps. Our manuscript provides support to these simple to use methods that give an estimate of soil carbon stock where estimates were previously largely non-existent, particularly at an individual saltmarsh level.*

Methods There is no mention of how the locations of transects and quadrats were chosen. Methods suggest that vegetation types were specifically chosen, but later (ln 118) it is mentioned that an analysis was conducted to determine how they fit in NVC classes. This sounds a bit circular. Were vegetation types specifically targeted?

*This section has now been altered to include the following text: 'The 4 vegetation types focused on in this study were located using governmental maps based on vegetation surveys from 1996-2003 (detailed in section 2.6). Vegetation type was therefore validated on the ground as species extent could have altered between the survey date and the present day.' (lines 115-118)*

It is not clear why the statistical analyses had to be restricted to a linear model (ln 150) – it should not be restricted to because citizen scientists might use it- application of models is not commonly tasks that citizen scientists perform. If so, authors could provide a spreadsheet to perform the calculation.

*Mixed effects models were chosen as they allow the estimation of fixed and random effects on the response variable (in this case surface SOC stock). The use of linear models is widespread. They usually assume a linear relationship between x and y variables (although you can always use transformations to model non-linear relationships using the same techniques, where needed [not in this case]). In addition, the analysis of variance that typically follows model building allow presenting the results in a format that is readily interpretable by members of both academia and governmental organisations. We have improved the text to make clear that the only thing that we ask citizen scientists to do as part of the Saltmarsh App is to check soil texture and vegetation type (lines 204-206). Members of the public are not required to carry out their own analysis at all.*

Location was divided into two classes, north and south Wales, and entered as a categorical variable. Is there a major biogeographical change between north and south? If latitude was considered important why not simply use latitude, rather than using a categorical value, to increase the ability to distinguish a gradient?

*Location was included as part of the model structure with site nested within location (north or south). If location is removed P value category (i.e. P < 0.001, P < 0.01 etc.) and r2 values (to 2 significant figures) remain the same, we therefore followed the reviewers suggestion and removed location from the model for the sake of simplicity.*

Vegetation covered To determine how geographically broad the results of this study could be one, needs to know more about the vegetation sampled, and that not sampled. Only 5 salt marsh vegetation classes are listed in this study – all simply identified by a dominant(?) species – two are identified by the same species, Juncus maritimus. How many quadrats were sampled in each vegetation class and were these equally distributed among the marshes?

*Part of this information is detailed in the supplementary material (Table S2). For the four vegetation types sampled, between 32 and 66 quadrats were surveyed, across a minimum of 9 and a maximum of 17 marshes, to reflect the dominant vegetation communities of the low, mid and high marsh saltmarsh communities along the coastline of Wales.*

 What proportion of cover is attributed to the dominant species? What types of species occur with the dominant? It would be useful to provide a table showing typical species composition and cover.

*This information is detailed in Rodwell (2000) and online at http://jncc.defra.gov.uk/pdf/Salt-marsh_Comms.pdf. Table 1 has also been edited (lines 147-150) to provide information on dominant and co-occurring species for each British National Vegetation Classification (NVC). Area coverage provided by the focus on 4 main vegetation types is two thirds or 66%, see line 398.*

It is likely that perennials will contribute more to soil carbon than annuals and graminoids over forbs (although Triglochin and Plantago can have substantial belowground biomass). Species richness was not found to be a significant explanatory variable, but what about the proportion of perennial vs annual plants? How many NVCs are there in UK salt marshes?

*While from a purely academic point of view this is a relevant point, we should bear in mind the aim of making our methods applicable for managers and non-academics. Hence, given that the proportion of perennial to annual plants are not mapped, the usefulness of this parameter usefulness in an applied tool such as the SCSP is limited. In contrast, NVCs are already mapped, what make them easy to use. There are 7 common NVCs found in the UK, 5 of which are considered in this manuscript. The two pioneer communities which were not assessed directly are included using mapped soil characteristics.*

Breadth of Geographical Application Authors suggest that their model can be used to estimate carbon stocks in the UK and perhaps northwestern Europe, as well. Yet, not all plant communities present in Welsh salt marshes were sampled (ln 289). What communities does this study miss from Wales and across the UK? How much salt marsh area is not accounted for? Authors further that their model could be applied from the Baltic to Portugal – is the vegetation really that consistent?

*Two pioneer saltmarsh communities common across Wales and the UK (Spartina and Salicornia), accounting for ~30% area of Welsh saltmarsh, were not directly assessed in this study, however their soil carbon stock is predicted on the basis of mapped simplified-soil characteristics. The 5 common saltmarsh vegetation communities focused on in this study are also widespread across north-western Europe.*

Unexplained variability Authors seem to have preliminarily truncated statistical analyses for this study. They note in the Discussion that _50% of the variation in the marshes they studied has yet to be statistically explained, further noting that the rest of the variation might be attributed to differences in grazing, salinity, pH, geomorpho-logical context, level of urbanisation, past disturbance, whether in a dynamic or stable area. Authors have reported data on grazing, salinity and pH that could easily be assessed in an expanded model. Geomorphological context can easily be determined from the maps in the supplementary material. As they mention "level of urbanisation" in the context of the study by Deegan et al. (2012). I assume they refer to nutrient loading of the estuary. Nutrient loading is not limited to urban development, but also to agricultural uses. If watershed nutrient loading models have been developed for UK estuaries the nutrient loading could be assessed as a predictor as well. Level of disturbance/ exposure seems to be similar to "whether the marsh sits in a dynamic or stable area", something that could be determined fairly easily.

*Wording in both the methods section (lines 168-171) and this section of the discussion (lines 423-427) have been improved as grazing, salinity and pH were indeed considered in early model selection but were not significant explanatory variables of soil carbon stock. Modelling geomorphological context, level of urbanisation and nutrient loading were considered beyond the scope of this manuscript which focuses on the prediction of soil carbon stock from simple measurements vegetation type and simplified soil type.*

Soil Carbon IPCC guidelines for calculation of greenhouse gas emission from land use change in coastal wetlands (Kennedy et al. 2013) suggest stocks be considered over 1 m depth. Granted such depths are difficult to sample and accurately measure bulk density, but not all soil samples in this study reached even 10 cm depth, yet this study is supposedly focussed on the upper 10 cm of soil. And, different soil parameters were measured over different depths.

*We have added a section to the Discussion (lines 384-395) to make clear that despite the fact that we are examining surface SOC stock, SOC stock in the top layer of soil is largely indicative of SOC stock in deeper soil layers. We will show that top SOC stock is indicative of deeper SOC stocks using data from Kingham, R.: The broad-scale impacts of livestock grazing on saltmarsh carbon stocks. PhD thesis, Bangor University, UK, 2013. This thesis includes 224 samples from a range of UK saltmarshes differing in soil type, plant community type and grazing intensity. By reanalysing this data set, we show that sampling to a depth of 10 cm consistently captures 72 ± 1 % of total soil organic carbon (Figure S5), when measuring to a depth of 45 cm. We thus argue that surface SOC stock can provide a reliable predictor of deeper carbon stores and is therefore a useful indicator of total SOC stock for UK saltmarshes.*

It is not clear how soil was sampled to determine bulk
density over 10 cm depth – and this is a very critical element, central to the entire
study. Text states that soil was collected from 2 cm to 9.5 cm. I suspect that the
sampling ring mentioned was not 3.1 cm high but 7.5 cm high (diameter and height
reversed in text?). Soil organic carbon was determined from this sample, as well. This
is not quite 10 cm and why was the surface 2 cm not collected?

*Text has been changed from 'vertically' to 'horizontally' (line 134) to improve the clarity of
explanation. In addition we have added a diagram to the supplementary material, Figure S2.
The bulk density core used was 3.1cm high and 7.5 cm in diameter, it was rotated into a
horizontal position to quantify the top 2 – 9.5 cm of soil in line with the methods used in Ford
et al. (2016) Soil stabilization linked to plant diversity and environmental context in coastal
wetlands. Excluding the top 2cm ensured that above ground vegetation was not accidentally
included within the bulk density core.*

The bulk density and
soil carbon measurements do not correspond to the soil texture which was determined
only on the surface 5 cm (ln 133). Do authors have any idea what the soil is like below
cm depth? Are any of the sampled marshes filled or previously drained ad now
restored?

*All of the saltmarshes in this study are semi-natural, i.e. they have not been filled or previously
drained but are often managed in some way (grazing livestock for food production or
conservation, right of way for coastal paths etc.). Soil texture was assessed by hand at ~5cm
depth to reflect the mid-point of the bulk density and soil carbon measurements (from a depth
of 2cm to 9.5 cm depth to quantify the top 10cm of soil). This method was used in Ford et al.
(2016) Soil stabilization linked to plant diversity and environmental context in coastal
wetlands.*

Did Emmett et al. (2010) establish a relationship between OC and LOI to
derive the conversion of 55% (Ln 131)?

*LOI values were compared to total soil C content measured by elemental analyser in Emmett
et al. (2010). Line 139.*

First National Inventory of blue carbon storage? It is a bit preliminary for authors to
claim to have the first national inventory of blue carbon storage.

*We have changed this to 'blue carbon storage in surface soil layers', lines 322-323. We would
like to emphasise that when we talk about National Inventory we do not mean UK-wide. We
are talking of a National Inventory, for Wales (a nation within the UK).*

Technical Editing

Figure 1 fig b needs a scale bar.

*A scale bar has been added to panel b.*

Figure 2 compares carbon stocks at a single marsh applying results of different models. However, because the areas covered are different it is not a fair comparison of the difference in carbon stocks predicted by the model.

*Figure 2 illustrates soil carbon stock in two ways, firstly surface SOC stock is shown visually (in t C ha$^{-1}$; 0-10 cm) using a grey scale with each section of marsh assigned a shade of grey based on predicted category of soil carbon stock, this scale is used regardless of saltmarsh area and matches the four models selected in this paper for use in the SCSP tool and the Saltmarsh App. Secondly each figure presents the area of the marsh in hectares alongside the total surface SOC carbon stock (in the top 10 cm of soil) for that area in t C. Panel D illustrates 'best practice', where the NVC_soil_model is used where NVC communities are mapped, with Soil_model used for the remaining saltmarsh area when NVC information is not available (not mapped), thereby giving the best estimate of soil carbon stock for the whole marsh area.*

Ln 41 Soil organic carbon IS belowground

*This sentence has been altered to avoid confusion (lines 41-43).*

Ln 45 I am surprised that salt marshes are considered terrestrial habitats

*In the Scholefield (2013) document they were classified as Coastal Margins alongside terrestrial habitats such as woodlands and grasslands. The text has been altered to reflect this (line 50).*

Ln 87 What current inventory?

*The wording in this sentence is a little misleading, it has now been altered for clarity (line 88). We refer to the current inventory as information compiled by the IPCC (2014) on blue carbon storage.*

Ln121 samples are dried to there is no longer a loss of moisture rather than for a prescribed time –Did authors assess whether 72 hrs adequate?

*Drying at 60 or 70 °C for 72 hours is a commonly used methodology for drying plant vegetation. We always test the amount of time needed to get constant weight. In our experience, with the amount of plant material we use it rarely exceeds 72 hours. This was tested with a subset of samples at the beginning of this study and in this case, 72 hours was enough for the small amount of above-ground biomass analysed (<5 g fresh weight).*

 Ln 367 what is meant by a "pioneer community" here?

*Pioneer communities are defined in the previous sentence as Spartina and Salicornia communities.*

Ln 390 Since level of disturbance/exposure seems to be
similar to "whether the marsh sits in a dynamic or stable area" seem to be the same there is no reason to cite an unpublished manuscript.

*We have removed this reference from the manuscript.*

Ln 269 shouldn't 0.45 be 45%?

*Footnotes to tables were edited to ensure use of percentage (45%) is clear (lines 286 and 296).*

References cited Kennedy HA, Alongi DM, Karim A, Chen G, Chmura GL, Crooks S, Kairo JG, Liao B, Lin G. 2013. Chapter 4 Coastal Wetlands In: Supplement to the 2006 IPCC Guidelines on National Greenhouse Gas Inventories: Wetlands.

*We have double checked citation guidelines for this publication. The online citation page for this supplement gives the format we have already used (IPCC, 2014) so we have not changed it.*

**Associate Editor Decision: Reconsider after major revisions**

Two reviewers have now evaluated your manuscript, both acknowledge the merit of the manuscript but also offer critical yet constructive suggestions for improvement. Based on their evaluation and your author replies, I would encourage you to provide a thoroughly revised version of your ms, which will be re-evaluated before a decision can be made. In case you have any specific questions, do not hesitate to contact us.

*Thank you for your comments.*

In your author replies and the Figures that it contains, you refer to the PhD study of Kingham (2013) to demonstrate the link between surface OC and depth-integrated OC stocks. I do note that "100% OC accounted for" corresponds to 45 cm depth - hence if I understand well the Kingham (2013) study only looked at a sediment depth up to 45 cm ? Please clarify this in your replies/revision - as this may still not correspond to the total sediment OC stocks ?

*Thank you for your comment, we've now made it clear that 100% of the carbon accounted for refers to a depth of 45 cm by mentioning this fact in the response to reviewers, the supplement and the new discussion section of the paper itself.*

[revised manuscript text omitted]

---

## Author Response (AR2)

*Reply to comments*

The authors claim that SOC typically has a log-linear relationship with depth. This may be true for terrestrial soils, but certainly not for tidal wetlands. In many tidal wetlands, SOC density is relatively stable with depth (those are mostly organogenic systems). I know that there are reports from minerogenic systems that show marked declines in SOC with depth, but I am not aware of any typical/generalizable SOC patterns in tidal-wetland soils (by the way, you do not provide reference for the newly cited reports in your reference list). This line of reasoning should be avoided and the authors should instead only argue with the new data they inlcuded form the Kingham thesis because this seems to be applicable to their study system (also I find this data almost too interesting for just bringing it in the supplement - but your call!).

*Thank you for pointing out that the log-linear relationship of SOC with depth cannot be generalised to all blue carbon systems. We have altered the text so that this relationship refers to minerogenic saltmarshes, with a focus on the Kingham dataset for quantifying the relationship between SOC and depth in UK saltmarshes (and used in this paper). In the previous paragraph, lines 371-374, we point out that our results would not necessarily be applicable to organogenic salt marshes dominated by Spartina species and common in North America. Bai et al. (2016) is included in the reference list, Drake et al. (2010) has been added. The reference to Fourqurean et al. (2012) has been removed as this refers to a seagrass ecosystem.*

You removed considerations on grazing from your discussion. I see that you now worked with a model that included grazing, but grazing was not significant. I think this is interesting and should be briefly stressed in the discussion (since you also mention grazing in the introduction and the topic recently gained some more attention in the literature). The authors have already provided the relevant literature in their previous version (Davidson 2017 , Mueller 2017, ...) so this shouldn't be a problem.

*Thank you for this point. We have now added text to the Discussion (lines 328-333) to highlight the fact that grazing intensity (grazed vs. un-grazed) was not indicative of surface SOC stock in this study, despite being related, in part, to plant community type. For the Davidson et al. (2017) meta-analysis a relationship between grazing and SOC stock was found, but only for American marshes. As the focus here is on European marshes we have used this paper to argue that our results are in line with those of other European marshes (75 in this meta-analysis).*